# Assessing and optimizing the role of wind forcing and upper-ocean dynamics in marine pollution transport simulations using surface drifters in the Canary Current System

Álvaro Cubas<sup>1</sup>, Francisco Machín<sup>2\*</sup>, Daura Vega-Moreno<sup>3</sup>, Eugenio Fraile-Nuez<sup>4\*</sup> and Borja Aguiar-González<sup>5</sup>

<sup>1</sup>University of Las Palmas de Gran Canaria, Las Palmas de Gran Canaria, 35017, Spain

<sup>2</sup>Oceanografía Física y Geofísica Aplicada (OFYGA), IU-ECOAQUA, University of Las Palmas de Gran Canaria, 35017, Spain

<sup>3</sup>OpenPLAS Group, Chemistry Department, University of Las Palmas de Gran Canaria, Las Palmas de Gran Canaria, 35017, Spain

<sup>4</sup>Centro Oceanográfico de Canarias, Instituto Español de Oceanografía (IEO), Consejo Superior de Investigaciones Científicas (CSIC), Santa Cruz de Tenerife, 38180, Spain

<sup>5</sup>EOMAR, IU-ECOAQUA, University of Las Palmas de Gran Canaria, Las Palmas de Gran Canaria, 35017, Spain

10

5

\*Correspondence to: Francisco Machín (francisco.machin@ulpgc.es) and Eugenio Fraile-Nuez (eugenio.fraile@ieo.csic.es)

#### Abstract.

- This study investigates the sensitivity of undrogued drifter trajectory simulations in the Macaronesia region, selected for their similarity to the behavior of marine litter transport in the upper ocean. The research evaluates the influence of various physical processes, including advection schemes, horizontal dispersion, windage and Stokes drift. A total of 320 simulations were conducted, incorporating different combinations of these processes, the modeled trajectories were compared with real drifter data. The analysis demonstrated that the inclusion of windage and/or Stokes drift significantly improved the
- agreement between modeled and observed trajectories, particularly when windage factors (WDF) ranged from 2.5% to 5%. Horizontal dispersion exhibited minimal influence on the trajectories, indicating that turbulent diffusion had a limited effect under the study conditions. While both advection schemes (RK2 and RK4) produced comparable results, RK4 outperformed RK2 in scenarios involving pronounced mesoscale activity. This research highlights the relevance of using undrogued drifters to mimic marine litter transport and underscores the importance of incorporating windage and/or Stokes drift in
- trajectory simulations, particularly in regions like Macaronesia, where mesoscale processes play a critical role.

# 1 Introduction

Marine pollution by microplastics is arguably one of the most pressing environmental challenges of our time. Addressing this issue requires a multidisciplinary approach that integrates a physical, chemical, and biological perspectives along with

<sup>15</sup> 

economic, social, and political dimensions (McGlade et al., 2021). The perspective of physical oceanography plays a key
role in uncovering the pathways of marine pollutants, identifying their accumulation zones, and determining their residence times. This information is critical not only for understanding the impact on marine ecosystems but also for providing essential tools to manage environmental crises in real time.

Once the marine litter enters the environment, reliable tools for effectively tracking its movement remain lacking. Instead, its behavior can be simulated by deploying virtual particles that mimic the dynamics of actual marine litter under specific

- environmental conditions (Castro-Rosero et al., 2023; Declerck et al., 2019; Jalón-Rojas et al., 2019). Such Lagrangian simulations typically require the inclusion of several parameters and complex equations that account for the major physical processes governing ocean dynamics. A common approach to evaluate the significance of these parameters or processes is sensitivity analysis, which provides insights into how variations in inputs  $x=(x_1,x_2,...,x_n)$  affect the output y. Inputs may include ocean currents, wind stress, number of particles, grid resolution, dispersion and drag coefficients, among others,
  - while *y* might represent the position of the particles at a specific location and time or the length of the impacted shoreline. Both inputs and outputs depend on the specific study being conducted and the framework used to calculate the trajectories, such as TrackMPD (Jalón-Rojas et al., 2019), OceanParcels (Lange and Sebille, 2017), PaTATO (Fredj et al., 2016) or OpenDrift (Dagestad et al., 2018), among others.
  - Here, we present a sensitivity analysis in the Canary Current System, comparing simulated trajectories with observed surface drifter trajectories to identify the optimal parameter values that best capture the physical processes governing the marine pollution transport in the region.

The circulation pattern in this system is predominantly influenced by the general circulation of the North Atlantic subtropical gyre, particularly its eastern branch, the Canary Current. This equatorward flow interacts with coastal upwelling waters and geographical structures such as the Canary Islands archipelago. The Canary Current exhibits strong seasonality,

intensifying during spring and summer before shifting offshore in fall (Machín et al., 2006; Mason et al., 2011; Pérez-Hernández et al., 2013; Stramma and Siedler, 1988).

Additionally, an equatorward coastal upwelling jet, known as the Canary Upwelling Current, originates north of Cape Ghir (Pelegrí et al., 2006). This current intensifies during spring (Machín and Pelegrí, 2006) and flows along the coast during spring and summer, driven by the seasonal variability of the prevailing winds (Cropper et al., 2014; Pelegrí et al., 2005)

The presence of the Canary Archipelago, which interrupts the main flow of the Canary Current generates significant mesoscale activity. This includes vortex streets downstream of the islands, creating a consistent pathway for eddies, known as the Canary Eddy Corridor (Sangrà et al., 2009). Numerous upwelling filaments are also present, some of which are quasipermanent features (Arístegui et al., 1997; Barton et al., 1998).

Reversals in the main flow have been observed near the Canary-Coastal Transition Zone during late autumn and winter (Navarro-Pérez and Barton, 2001). These flow changes are likely caused by a weakening of the trade winds south of Cape Ghir (Pelegrí et al., 2005), which allows the development of a northward flow between Cape Blanc and Cape Juby (Hernández-Guerra et al., 2002; Machín and Pelegrí, 2009). In addition, the interaction of the trade winds with the islands

diverts airflow to the flanks, creating warm wakes and altering surface circulation in the lee regions (Barton et al., 2000; Basterretxea et al., 2002; Hernández Guerra, 1990).

Regarding wind patterns, the region between 20° and 30°N experiences year-round upwelling-favorable wind stress along the northwest African coast (Bakun and Nelson, 1991). Furthermore, the Canary Islands region (28°-29°N) is periodically affected by intermittent pulses of dust clouds from North Africa, known as *calima*, which peak mainly in winter and summer/autumn (Torres-Padrón et al., 2002).

The objectives of this research are to investigate the influence of multiple oceanographic and meteorological processes, such as wind forcing and Stokes drift, on the trajectories of surface drifters in the Canary Current System. This is achieved 75 by comparing observed trajectories with those modeled trajectories using TrackMPD. By conducting a sensitivity analysis, we aim to identify the optimal parameter values for the most accurate trajectory modeling. This approach not only clarifies the role of each process but also addresses challenges in parameter estimation, ultimately enhancing the replication of pollutant transport dynamics in the region.

#### 2 Material and Methods 80

# 2.1 Drifters data

We utilized satellite-tracked drifters from the hourly drifter data provided by the Global Drifter Program (GDP) (Elipot et al., 2016, 2022). These buoys, with a half-life of 1.5 to 2 years, employ Iridium SBD telemetry and are equipped with a 6-meter Holey sock drogue centered at 15 meters depth. The drifters are constructed using two hemispheres of acrylonitrile butadiene styrene (ABS), forming a spherical float. Additionally, they are fitted with strain gauges to detect whether the drogue remains attached. Detailed specifications for each selected drifter are provided in Table 1.

Table 1. Specifications of selected drifters. ID: GDP identification number; WMO: World Meteorological Organization identifier; SVPB: Surface Velocity Program Barometer; SVP: Surface Velocity Program.

| Name | ID              | WMO     | Experiment<br>Number | Buoy<br>Type | Diameter<br>(cm) | Manufacturer | Manufacture<br>Year |
|------|-----------------|---------|----------------------|--------------|------------------|--------------|---------------------|
| TR1  | 300234065704790 | 4101609 | 21321                | SVPB         | 35.5             | PacificGyre  | 2016                |
| TR2  | 300234065749810 | 4101717 | 21321                | SVPB         | 40               | MetOcean     | 2016                |
| TR3  | 300234066214360 | 6203843 | 21421                | SVPB         | 35.5             | Pacific Gyre | 2020                |
| TR4  | 300234066797700 | 4401580 | 2222                 | SVPB         | 40               | MetOcean     | 2019                |
| TR5  | 300234066892260 | 6203634 | 2222                 | SVPB         | 40               | MetOcean     | 2019                |
| TR6  | 300234067973120 | 6203776 | 21321                | SVPB         | 35.5             | Pacific Gyre | 2020                |

| Name | ID              | WMO     | Experiment<br>Number | Buoy<br>Type | Diameter<br>(cm) | Manufacturer | Manufacture<br>Year |
|------|-----------------|---------|----------------------|--------------|------------------|--------------|---------------------|
| TR7  | 300234068245900 | 6203597 | 21312                | SVP          | 38.1             | SIO          | 2019                |
| TR8  | 300534060651930 | 4402660 | 21312                | SVPB         | 39               | DBi          | 2020                |
| TR9  | 300534061170370 | 6204561 | 2222                 | SVPB         | 40               | NKE          | 2020                |
| TR10 | 300534061655130 | 1301717 | 21421                | SVPB         | 35.5             | Pacific Gyre | 2020                |

- We selected drifters with trajectories passing near the Canary Islands and Madeira archipelagos, within a latitude range of 27°N to 34°N and a longitude range of 19°W to 8°W, over the temporal domain off 2021 to 2022. The data were further filtered to include only trajectories where the drogue had already been lost, ensuring proper consideration of windage effects (Brügge and Dengg, 1991; Pazan, 1996). From this dataset, we identified 10 trajectories (*TR<sub>i</sub>*), to capture a variety of physical processes that could either hinder or facilitate the trajectory computation (Figure 1). To assist in the selection, we computed key trajectory characteristics such as duration, effective distance, track stability (the ratio of effective distance to
- total distance), and others (Table 2).

Figure 1. Trajectories of the selected undrogued drifters (TR) used in the sensitivity analysis.