# Peer review of "Assessing and optimizing the role of wind forcing and upper-ocean dynamics in marine pollution transport simulations using surface drifters in the Canary Current System"

_EGUsphere, 2024_

## Author Comment (AC1)

**Dear Referee #2,**

We sincerely appreciate your comments on our manuscript, *"Assessing and optimizing the role of wind forcing and upper-ocean dynamics in marine pollution transport simulations using surface drifters in the Canary Current System."* Your feedback is highly valuable and has helped us identify areas for improvement. Below, we address each of the points raised in detail:

1. **Lack of novelty and substantial conclusions**

   o We agree that many of the results confirm what is already known from previous studies. However, our goal was to conduct this sensitivity study specifically in the Canary region, setting a precedent for applying these types of tools in emergency situations in the area. We understand the point raised by the reviewer, and we will take it into account in future work to achieve a greater degree of innovation.

   o Nevertheless, we present our results in the context of emergency response exercises in Macaronesia, which we consider a novel application for this region.

2. **Diffusivity coefficients and lack of discussion**

   o The diffusivity values presented were chosen based on similar studies, such as Jalon-Rojas et al. (2019), and we attempted to discuss them in the context of the presence of nonlinear phenomena. However, we understand the comment by the reviewer regarding the lack of discussion on why the Skill Score should increase and the need for better justification of the chosen values.

3. **Differences between numerical schemes and time step**

   o We agree with this comment, and we have not considered it, but we greatly appreciate it. It could indeed be interesting to take this into account for future work, especially when using different time steps or an adaptive time step.

4. **Figure 2 presentation**

   o We understand the difficulties presented by Figure 2 due to its dependence on Table 3. We have not found a way to correct this without creating a visual overload. Any suggestions on how to improve this are welcome, and we would be open to hearing them to enhance future work. Thank you very much.

**Conclusion**

Thank you again for your detailed review. We appreciate your feedback and will take it into account in future work.

---

## Author Comment (AC2)

**Dear Referee #1,**

We sincerely thank you for your comments on our manuscript titled *"Assessing and optimizing the role of wind forcing and upper-ocean dynamics in marine pollution transport simulations using surface drifters in the Canary Current System."* Your observations are highly valuable and have helped us identify areas for improvement. Below, we address each of the points raised in detail:

1. **Limited drifter dataset and selection of the study period**

   o We appreciate the comment raised by the reviewer as it gives the opportunity to explain in more detail the constrains faced with the databases selected. We have used the available data temporal domain since we were constrained on one hand by the hydrodynamic data and on the other by the availability of drifter data. For this reason, given that the IBI Analysis and Forecast model regularly removes its older data, the amount of available data today would be even smaller than when the experiments presented were conducted. In any case, thank you for your comment; we will take it into account in future experiments.

2. **Information on the Lagrangian modeling framework "TrackMPD"**

   o We believe that we have provided sufficient details about TrackMPD adding also a reference to the manuscript by its developers in case the reader would be interested in more details about this Lagrangian model.

   o Regarding the differences between TrackMPD and other tools, there are no significant differences in the 2D model. However, when considering a third vertical dimension, the differences become more substantial. For more information on these differences between TrackMPD and similar tools, you can refer to the following article: Bigdeli et al. (2022) (https://doi.org/10.3390/jmse10040481).

3. **Lack of discussion in the results**

   o It is possible that the structure we chose was not the most appropriate, as combining the results and discussion may have diluted the discussion within the results. Thank you for your comment; we will take it into account for future work.

4. **Insufficient references of previous drifter-based studies**

   o We believe we have included several references based on studies of a similar nature. In particular, we referenced two manuscripts about the Global Drifter program in the Data section and some more within the Results and discussion section. We would be grateful to know the references missed by the reviewer.

5. **Lack of a map of the study area**

   o We understand the need to present a figure representing the circulation system of the region to provide context for the observations, and we will take it into consideration in future projects.

**Conclusion**

Thank you again for your review. We appreciate the comments and will take them into consideration for future work.

---

## Author Comment (AC3)

Dear Dr. van Sebille,

Thank you for your email and for the opportunity to have our manuscript considered for publication in Ocean Science. We appreciate the time and effort that both you and the reviewers have invested in evaluating our work.

We have carefully reviewed the referee comments and acknowledge their critical assessment regarding the novelty and relevance of our study, among other comments. While we regret that our manuscript does not align with the scope and expectations of Ocean Science, we recognize the value of the feedback provided. We will take these comments into account as we refine our study for submission to a more suitable journal.

Once again, we appreciate the constructive critique and the editorial handling of our submission. We hope to have the opportunity to contribute to Ocean Science in the future.

Best regards,

Álvaro Cubas on behalf of all coauthors.